# Identification of Sorghum (*Sorghum bicolor* (L.) Moench) Genotypes with Potential for Hydric and Heat Stress Tolerance in Northeastern Mexico

**DOI:** 10.3390/plants10112265

**Published:** 2021-10-22

**Authors:** Marisol Galicia-Juárez, Francisco Zavala-García, Sugey Ramona Sinagawa-García, Adriana Gutiérrez-Diez, Héctor Williams-Alanís, María Eugenia Cisneros-López, Raúl Enrique Valle-Gough, Rodrigo Flores-Garivay, Jesús Santillano-Cázares

**Affiliations:** 1Instituto de Ciencias Agrícolas, Universidad Autónoma de Baja California, Mexicali 21705, Baja California, Mexico; marisol.galicia.juarez@uabc.edu.mx (M.G.-J.); raul.valle@uabc.edu.mx (R.E.V.-G.); rodrigo.flores.garivay@uabc.edu.mx (R.F.-G.); 2Facultad de Agronomía, Universidad Autónoma de Nuevo León, General Escobedo 66050, Nuevo León, Mexico; francisco.zavalag@uanl.mx (F.Z.-G.); sugey.sinagawagr@uanl.edu.mx (S.R.S.-G.); adrina.gutierrezdz@uanl.edu.mx (A.G.-D.); 3Instituto Nacional de Investigaciones Forestales, Agrícolas y Pecuarias, (INIFAP), Río Bravo 88900, Tamaulipas, Mexico; hector.williamsa@uanl.mx (H.W.-A.); cisneros.maria@inifap.gob.mx (M.E.C.-L.)

**Keywords:** relative water content, damage to cell membrane, photosystem II, maximum quantum yield (Fv/Fn), sorghum

## Abstract

Sorghum (*Sorghum bicolor* (L.) Moench) is cultivated in regions with frequent drought periods and high temperatures, conditions that have intensified in the last decades. One of the most important photosynthetic components, sensible to hydric stress, is maximum quantum yield for photosystem II (PSII, or Fv/Fm). The objective of the present study was to identify sorghum genotypes with tolerance to hydric and heat stress. The treatments were hydric status (hydric stress or non-hydric stress (irrigation)), the plant’s developmental stages (pre or post-anthesis), and six genotypes. The response variables were Fv/Fm; photosynthetic rate (P_N_); stomatal conductance (g_s_); transpiration rate (E); relative water content (RWC); damage to cell membrane (DCM) at temperatures of 40 and 45 °C; and agronomic variables. The experiment was conducted in pots in open sky in Marín, N.L., in the dry and hot northeast Mexico. The treatment design was a split–split plot design, with three factors. Hydric stress diminished the functioning of the photosynthetic apparatus by 63%, due to damage caused to PSII. Pre-anthesis was the most vulnerable stage to hydric stress as it decreased the weight of grains per panicle (85%), number of grains per panicle (69%), and weight of 100 grains (46%). Genotypes LER 1 and LER 2 were identified as tolerant to hydric stress, as they had lower damage to PSII; LER 1 and LEB 2 for their superior RWC; and LER 1 as a thermo tolerant genotype, due to its lower DCM at 45 °C. It was concluded that LER 1 could have the potential for both hydric and heat stress tolerance in the arid northeast Mexico.

## 1. Introduction

In many regions of the world, drought and high temperatures occur simultaneously. However, often, they are addressed independently. High temperatures interact with drought to affect the hydric relations of crops [1,2,3,4], especially in northeast Mexico, which is an important agricultural region where cereals are grown extensively [5]. Sorghum (*Sorghum bicolor* (L.) Moench) is one of the most important cereals in the world in semiarid and subtropical regions [6,7,8]. Although sorghum is a crop adapted to dry climates, hydric stress is considered the most important abiotic limiting factor for growth and productivity [9,10,11,12]. This negative effect can be mitigated through genetic improvement, taking advantage of the genetic variability of the species through tolerant germplasm evaluation and selection [13,14,15]. To achieve such goal, is important to understand the physiological responses caused by drought and high temperatures [1,10,16,17].

The cell membrane stability and the relative water content (RWC) in leaves are physiological selection indices in plants when selecting for tolerance to hydric stress or high temperatures [18,19,20,21,22,23]. Hydric stress affects the water metabolism in plants, as it limits nutrient absorption [24,25,26], transpiration rates, and plant growth [4,27,28] and interferes with physiological and biochemical processes which diminishes yield and crop quality [15,29]. Additionally, hydric stress drastically diminishes growth parameters such as biomass accumulation, root to shoot ratio, foliar area, and chlorophyll concentration [30,31,32,33].

Photosynthesis is the most affected physiological process caused by hydric stress due to cell turgor pressure loss, which, in turn, reduces the maximum carboxylation rate, causes damage to ATP production, and increases mesophyll resistance to CO_2_ diffusion, resulting in stomatal closure and reduced mesophyll conductance [34,35]. Reddy et al. [36] provided an ample analysis on biochemical and molecular plant responses to drought. Hydric stress also inhibits electron transport rates, which increases oxidative stress and can seriously affect the photosynthetic apparatus of leaves [37,38,39]. PSII is located in the thylakoids membranes of mesophyll cells and is considered the most sensible component to drought stress and also the main source of fluorescence [38,40,41]. 

The maximum quantum yield of PSII (Fv/Fm), expressed by the maximum chlorophyll florescence (Chlorophyll a, Chl a), is considered one of the most important photosynthetic components [42,43]. The maximum chlorophyll fluorescence of Chl a in photosystem I is weak and can be expected to be constant; while in photosystem II (PSII), fluorescence of Chl a is strong and varies over time as light intensity and water supplies change. Therefore, by measuring the fluorescence of Chl a, both maximum quantum yield for PSII and thermal energy dissipation in the antenna of PSII can be estimated [44,45]. Florescence measurements of Chl a has been used in photosynthesis research, as it is possible to obtain detailed information about the photosynthetic apparatus [44,45,46]. The hypothesis of the present research was that there is at least one genotype that will shows tolerance to hydric and heat stress, from a group of promissory experimental lines that have been selected by two Mexican breeding programs. 

The grain sorghum genotypes consisted of three experimental lines B (LEB) (maintainers); two experimental lines R (LER) (restorers); and one commercial hybrid as control (King Gold^®^); which has demonstrated tolerance to hydric and to high temperatures stresses in the state of Nuevo Leon, Mexico [47]. LER 1 is an R line selected for drought and heat tolerance in the Facultad de Agronomía, Universidad Autónoma de Nuevo León (in the dry and hot northeast Mexico). LER 2 is also an R line, but this was selected by the Colegio de Posgraduados (high elevation, close to Mexico City), with adaptation to cool weathers. LEB 1, LEB 2, and LEB 3 (self-pollinated) were also selected by the Colegio de Posgraduados [48]. The LEB are insensible to photoperiod; their limbs show no anthocyanin pigmentation; tannins in grain are scarce or absent; and they exhibit floury predominant endosperm (75%), and light brownish glumes. The color of the grain is ‘A’/‘B’, creamy and opaque [48]. It was of interest testing the lines selected in the Colegio de Posgraduados under highly contracting weather conditions of northeast Mexico. The objective of the present study was to identify sorghum genotypes, with potential for drought and or heat stress tolerance, through assessing the Fv/Fm of PSII, physiological, and agronomical responses.

## 2. Results

### 2.1. Ambient Temperatures and Soil Humidity

Recordings of ambient temperatures during the hydric stress application period pre- anthesis (28 days) ranged from a minimum of 17 to 25 °C and a maximum of 25 to 40 °C; while at the post-anthesis stage (14 days), the minimum temperatures ranged from 21 to 25 °C and the maximum from 36 to 40 °C (Figure 1). Student’s *t*-test for soil humidity recorded a significant difference at the pre-anthesis stage, with mean values of 91.2% under irrigation and 70.3% under hydric stress; post-anthesis, soil humidity was also significant among treatments, recording mean values of 84.7% under irrigation and 76.4% under hydric stress. Soil humidity variation across sampling days pre- and post-anthesis are shown for both irrigation and humidity stress treatments (Figure 2).

### 2.2. Physiological Variables

#### 2.2.1. Maximum Quantum Yield of PSII (Fv/Fm) under Hydric Stress

Table 1 shows the significance of Fv/Fm for the soil humidity–genotype interaction. This interaction occurred as, under irrigation, no significant differences were recorded among genotypes. However, under hydric stress, the commercial hybrid decreased by 36%, i.e., a drop in the functioning of the photosynthetic apparatus (electron flux, specifically), in relation to Fm/Fv under irrigation (Figure 3). Among experimental lines, similar magnitudes of reductions in photosynthetic functioning were observed for LER 1 and LER 2, which had reductions of 48 and 49%, respectively, but LEB 2, LEB 1, and LEB 3 had very different results, which showed reductions of 80, 81, and 83%, respectively.

#### 2.2.2. Photosynthetic Rate (P_N_), Stomatal Conductance (g_s_), and Transpiration (E)

P_N_, g_s_, and E were all mostly affected by soil humidity treatments (see mean square magnitudes and significance value in Table 1), although g_s_ also recorded significant soil humidity–stage interaction. This interaction occurred as, in post-anthesis, both water regimes yielded very similar g_s_ mean values, with 0.30 and 0.29 mol H_2_O m^−2^ s^−1^ for irrigation and for hydric stress, respectively. Pre-anthesis, the difference was more than 100%, with 0.26 and 0.11 mol H_2_O m^−2^ s^−1^ under irrigation and under hydric stress, respectively. E was significantly affected by soil humidity and by stage. E recorded 8.9 and 4.3 mmol H_2_O m^−2^ s^−1^ under irrigation and hydric stress, respectively. By stages, the mean values were 8.62 and 10.66 mmol H_2_O m^−2^ s^−1^ pre- and post-anthesis, respectively. Table 2 shows how hydric stress caused these three photosynthetic related variables (P_N_, g_s_, and E) to dramatically decrease, as compared with irrigation. Means in Table 2 are averaged over genotypes, as P_N_, g_s_, and E were not affected by genotypes.

#### 2.2.3. Relative Water Content (RWC)

Relative water content was significantly affected by genotypes and soil humidity (Table 1). Figure 4 shows the RWC of all six genotypes. RWC is directly proportional to soil water availability.

#### 2.2.4. Damage to Cell Membranes at 40 and 45 °C

No main factor or any of the interactions affected damage to cell membrane at 40 °C (Table 1). However, a significant soil humidity–genotype interaction was observed for damage to cell membrane at 45 °C (Table 1). This interaction occurred as genotypes LER 1, LEB 3, LEB 1, and the control had similar damage to cell membrane at 45 °C at both soil humidity regimes, while genotypes LEB 2 and LER 2 significantly differed in damage to cell membrane at 45 °C when tested under the two soil humidity regimes (Figure 5).

### 2.3. Agronomic Variables

#### 2.3.1. Days to Flowering

Days to flowering was affected by the most complex soil humidity–stage–genotype interaction (Table 3). This three-way interaction resulted due to the unique performance of genotype LER 2 (Figure 6). Four out of five treatments sharing the letter a (the latest flowering treatments) occurred when the hydric stress was applied pre-anthesis. This was observed in the control, LEB 3, LER 1, and LER 2, but the exception was LER 2, which also flowered latterly, but did so under both irrigation and under hydric stress applied post-anthesis (Figure 6). For this reason, also significant were the second order stage–genotype and soil humidity–stage interactions (Table 4).

#### 2.3.2. Plant Height

Plant height recorded a soil humidity–stage interaction (Table 3). This interaction resulted as plant heights were very even across soil humidity treatments pre- and post-anthesis when hydric stress was applied post-anthesis, but the treatment of hydric stress applied pre-anthesis resulted in significantly shorter plants (Table 4). The difference between plant height under irrigation and pre-anthesis was 40% higher than plants under hydric stress at the same stage (pre-anthesis) (Table 4).

#### 2.3.3. Panicle Length

Panicle length also recorded a soil humidity–stage interaction (Table 3). Panicle lengths were very uniform across treatments pre-anthesis under irrigation and post-anthesis under hydric stress and irrigation, but significantly different from pre-anthesis under hydric stress (Table 4). Genotype was also significant for panicle length, with 25, 25, 25, 23, 23, and 22 cm, for the control, LEB 3, LEB 2, LER 1, LEB 1, and LER 2, respectively. The control, LEB 3, and LEB 2 formed the group with the largest panicles, and the remaining genotypes formed another group with significantly shorter panicle lengths. Plant height and panicle length reached their maximum values before post-anthesis; thus, these variables were unaffected by soil humidity post-anthesis (Table 4).

#### 2.3.4. Grain Weight per Panicle (GWP), Weight of 100 Grains (W100G), and Number of Grains per Panicle (NGP)

Plant height and panicle length, GWP and W100G, also recorded a soil humidity–stage interaction (Table 3). Furthermore, they performed identically in response to treatments, recording lower means of hydric stress than treatments under irrigation, independent of the stage at which the stress occurred. However, the interaction occurred as, among both stages under hydric stress, the one where the hydric stress was imposed pre-anthesis was drastically lower than when the hydric stress was applied post-anthesis, particularly for grain weight per panicle (Table 4). GNP was only significantly affected by soil humidity (Table 3), recording 2846 and 995 grains per panicle, under irrigation and under hydric stress, respectively (almost 300% difference in favor of irrigation).

## 3. Discussion

### 3.1. Physiological Variables

#### 3.1.1. Maximum Quantum Yield of PSII (Fv/Fm) under Hydric Stress

The mean Fv/Fm for soil humidity was 0.76 under irrigation and 0.28 under hydric stress. This difference represents 37%. Under hydric stress, the range of mean values was from 0.13 to 0.49. Such values indicate damage to PSII, likely caused by damage to thylakoid membranes; a greater production of oxygen reactive species or increased membrane permeability, which likely caused a proton-leaking and, thus, a decrease in ATP and NADPH production; and a reduction in photochemical efficiency [49,50]. Genotypes under irrigation, on the other hand, recorded a Fv/Fm range of 0.70 to 0.78, which falls within the category of healthy leaves [43]. In leaves not under hydric stress, Bilger et al. [43] recorded values of 0.75 to 0.85. Netondo et al. [51] reported a Fv/Fm value of around 0.78–0.79, for salinity stress free sorghum leaves, falling to about 0.72 for the highest salt concentration in two sorghum varieties.

It is important to notice that, in the present study, the lowest Fm/Fv mean (0.28) was recorded for the hydric stress treatment, due to the occurrence of temperatures close to 40 °C during post-anthesis, which caused a decrease in photosynthetic rate. Qasem et al. [52] suggested that hydric stress causes changes in plant respiration, stomatal closure, high foliar temperature, chlorophyll disintegration, and damage to PSII, in the photosynthetic apparatus. According to Kapanigowda et al. [50], post-anthesis hydric stress and significantly increased chlorophyll loss led to decreases in Fv/Fm.

Under the assumption that Fv/Fm would be less affected during hydric stress in tolerant genotypes, the occurrence of the soil humidity–genotype interaction allowed the identification of hydric stress tolerant genotypes. Except for the commercial hybrid, such genotypes were LER 1 and LER 2.

#### 3.1.2. Photosynthetic Rate (P_N_), Stomatal Conductance (g_s_), and Transpiration (E)

Results similar to those found in the present study were reported by Elsheery and Cao [53], who suggested that that drought reduces P_N_, which would result from stomata closure and thus a reduction in CO_2_ diffusion to mesophyll cells. Akman et al. [15] reported P_N_ rates of 2.8 to 8 µmol m^−2^ for water stress applied pre-anthesis and 3 to 9.9 µmol m^−2^ post-anthesis. Considering that in this study the value for hydric stress (averaged across the two phenological stages) was as low as −1.9 µmol m^−2^, it is indicative of the severity of the stress the plants suffered under the hydric stress treatment.

Other kind of stresses, such as soil salinity, heat, and cold negatively affect CO_2_ assimilation, where low g_s_ and damage to the photosynthetic apparatus are considered as key factor in reducing CO_2_ assimilation [54,55]. Stomatal closure causes an increment in surface temperatures of plant leaves, and the magnitude of increment depending mainly on solar radiation intensity over the plants’ canopies, but would translate in water savings or in an increment of transpiration efficiency [56].

Under water availability conditions, stomatal regulation keeps an optimum internal CO_2_ concentration level that allows the plant to fulfill the CO_2_ demand for the Calvin cycle to continue. However, under hydric stress conditions, there will be a compromise between the plant’s need to keep a functional hydric balance and the need of producing carbohydrates [57,58]. Under irrigation conditions for wheat and other crops, positive relations have been recorded between grain production and g_s_ [59,60,61]. Condon et al. [62] measured leaves porosity in experimental lines for improvement under irrigation conditions, finding that g_s_ pre- and post-anthesis explained 50 and 65% of grain yield variance, respectively. Under irrigation environments, a high g_s_ indicates a high E, and biomass accumulation, as a result.

#### 3.1.3. Relative Water Content (RWC)

RWC results in the present study are in agreement with Mahfouz et al. [63]. By comparing the mean values under hydric stress (32.5%) versus the non-stressed plants (85.3%); the difference account to a decrease of 63% on RWC. It has been demonstrated that RWC in leaves decreases with increasing hydric stress [63]. A decrement in RWC is one of the first symptoms of water deficiency in leaves; thus, this is considered the most significant character when identifying differences among genotypes [64]. These results allowed us to identify LER 1 and LEB 2 as tolerant genotypes, as these recorded the highest values of RWC, which were similar to the control genotype.

#### 3.1.4. Damage to Cell Membranes at 40 and 45 °C

This analysis allows the identification of thermo tolerant genotypes; as the cell membrane stability is one secondary (or indirect) character, useful to study the effects of hydric and heat stress, as it is a quantitative trait that is moderately heritable and with a high genetic correlation with grain yield [65]. Genotypes were considered as thermo tolerant when damage to cell membrane at 45 °C mean values under hydric stress were similar (or lower) than the same genotypes under irrigation. Therefore, LER 1 was identified as thermo tolerant as it recorded a similar damage to cell membranes at 45 °C, regardless of soil water status (the two bars of LER 1 are almost the same length in Figure 3).

Damage to cell membrane at 45 °C was also affected by soil humidity regimes. Damage to cell membrane at 45 °C recorded the highest damage under hydric stress, recording a mean of 81.2%. As this kind of stress caused damage to cellular membrane, making it more permeable to various ions and metabolites, it probably caused the scape of electrolytes [63]. Furthermore, the high temperatures occurring post-anthesis could have favored a collapse in cell organization and functioning. Such damage would include denaturalization and protein aggregation and can result in a more fluid lipid membrane [66].

### 3.2. Agronomic Variables

#### 3.2.1. Days to Flowering

Pre-anthesis, and under hydric stress, plants flowered at 74 days. Hydric stress caused a delay of eight days, compared with irrigation (66 days). However, post-anthesis, plants flowered almost at the same time, regardless of soil hydric condition. Even in some genotypes such as the control, LER 1, LER 2, and LEB 2, an interruption of the emergence of the panicle was observed. Similar results were reported by Craufurd and Peacock [11], where the appearance of the panicle delayed two to 25 days and flowering about 59 days; thus, fully inhibiting panicle development in some genotypes, due to severe hydric stress. In the post-anthesis stage, hydric stress treatments did not show statistical differences on days to flowering (all sharing the letter b, Figure 6); as hydric stress was applied once flowering had occurred.

#### 3.2.2. Plant Height

Just as discussed for days to flowering, plant height was only affected pre-anthesis, producing shorter plants under hydric stress than under irrigation. Plants’ height was unaffected post-anthesis, independently of soil hydric condition (as plant height had stopped at the beginning of the reproductive stage). Plant height decrement caused by stress applied pre-anthesis in the present study coincide with results found by Akman et al. [15] who reported a decrease in plant height when hydric stress was applied pre-anthesis. They reported plant heights pre- and post-anthesis ranged from 49 to 107.7 cm and 74 to 105.3 cm, respectively.

#### 3.2.3. Panicle Length

Both plant height and panicle length decrease pre-anthesis under hydric stress mainly as these plant characters define their expression during this phenological stage, which likely induced a decrease in cell turgidity, obstructing water flux from the xylem to adjacent phloem cells, inhibiting their development, and ultimately stopping their growth [67]. Although, in the present study, the magnitude of shortening panicle lengths when hydric stress was applied pre-anthesis was very clear, with a 92% difference (26.7 and 13.9 cm, from Table 4), these results are in agreement with other similar research. Akman et al. [15] reported that panicle length in plants grown under irrigation and drought conditions were shorter when the hydric stress was applied pre-anthesis (19.9 cm, the mean of three genotypes) than when the stress was applied post-anthesis (22.4 cm, the mean of three same genotypes); a 13% difference. However, panicle biomass, in Aikman’s report [15], indicates a very large difference between applying hydric stress pre- and post-anthesis, with means of 12.8 and 24.8 gr, respectively (a 93% difference), suggesting, as bottom line, that water stress pre-anthesis severely reduces panicle size.

#### 3.2.4. Grain Weight per Panicle, Weight of 100 Grains, and Number of Grains per Panicle

The low grain weight per panicle values is attributed mainly to the number of grains per panicle, which was severely affected by hydric stress at both stages (Table 4) which, compared with the same genotypes under irrigation, recorded a decrement of 71% pre-anthesis, and 67% post-anthesis. Furthermore, hydric stress caused a reduction in the weight of 100 grains of 49% pre-anthesis, and 27% post-anthesis. The lowest values were recorded during pre-anthesis, which is in agreement with Reddy et al. [68]; who stated that, during this stage, hydric stress (drought) causes a drastic reduction in grain yield as compared with hydric stress post-anthesis. Number of grains per panicle recorded 2846 grains per panicle under irrigation and 995 grains per panicle under hydric stress. This represents a difference of 286% in favor of irrigation. Jabereldar et al. [69] reported a decrease in number of grains per panicle when water stress was applied at eight leaves stage than at three leaves stage in two growing seasons, with 1323 and 1577, and 1265 and 1385, respectively. These results are similar to those found in sorghum under hydric stress conditions at stages pre and post-flowering [70,71].

## 4. Materials and Methods

### 4.1. Experimental Location

The experiment was conducted at the FAUANL experimental station, at Marin, Nuevo Leon, Mexico, located at 25°52′ N–100°02′ W, at an elevation of 355 m above the sea level. The climate corresponds with BSL (h) w (e), which is described as dry warm steppe, with summer rains [72]. The location has an average annual precipitation of 595 mm and an average annual temperature of 22 °C.

### 4.2. Experimental Conditions

This experiment was run using pots of 25 cm diameter and 55 cm depth, in open sky conditions. The seeding date was 20 of March, 2019, in the spring–summer growing cycle. In every pot of soil, humidity was measured every four days using a soil moisture tensiometer AQUATERR^®^, model EC-350 (Costa Mesa, CA, USA). Air temperature was recorded with a data logger Extech^®^, model TH10 (Burlington, VT). Six sorghum genotypes were seeded in pots containing a mixture of sandy-clayey soil (3.9% organic matter and 0.35% nitrogen (N), pH of 7.9). No other nutrient application was made throughout de duration of the experiment. Three seeds were seeded in the center of each pot at a depth of 3 cm. After 10 days, two plants were removed to leave only one plant per pot. Thus, one plant per pot was considered as experimental unit.

### 4.3. Experimental Design

The experiment was arranged as a split–split plot in a completely randomized design, with four replications. The whole plots were two developmental stages (pre-anthesis or post-anthesis); the sub-plots were assigned to two soil humidity treatments (irrigated or hydric stress); and six grain sorghum genotypes were the sub-sub plots (Control (Commercial hybrid King Gold^®^). LEB 1. LEB 2, LEB 3, LER 1, and LER 2) (Figure 7). At the pre-anthesis stage, the hydric stress consisted of restricting irrigation 40 days after the seeding date, until 64 days after the seeding date. At the post-anthesis stage, the hydric stress started when the average of all plants were at flowering, 64 days after the seeding date.

### 4.4. Physiological Variables

Fv/Fm measurements and physiological variables started 77 days after the seeding date. Fv/Fm was measured on the flag leaf using a fluorometer Li-6400 (LI-COR Inc., Lincoln, NE, USA) with an integrated fluorescence chamber head (Li-6400-40; LI-COR Inc., Lincoln, NE, USA). The flag leaf of each plant was previously adapted to darkness by covering a section with aluminum foil for 20 min. After that, the aluminum foil was removed, and the readings were taken. Furthermore, during measurements, a black plastic film was put over both the entire leaf and the fluorometer chamber, in order to block any illumination reaching the darkness acclimated leaf. *Fv*/*Fm* was computed as follows:(1)$$\frac{Fv}{Fm}=\frac{Fm-F0}{Fm}$$
where: *F*0 = Minimum leaf chlorophyll fluorescence after dark acclimation*Fm* = Maximum leaf chlorophyll fluorescence after dark acclimation*Fv* = Variable leaf chlorophyll fluorescence after dark acclimation

Photosynthesis and gas exchange variables were measured with a Li-6400 apparatus. Net photosynthetic rate (P_N_, µmol CO_2_ m^−2^ s^−1^), stomatal conductance (g_s_, mol H_2_O m^−2^ s^−1^), and transpiration rate (E, mmol H_2_O m^−2^ s^−1^). Calibration conditions for the Li-6400 during the measurements were as follows: Flow rate of 700 µmol s^−1^, a constant level of CO_2_ of 400 µmol CO_2_, block temperature of 25 °C, and a light source LED 6400-02 at 1500 µmol m^−2^ s^−1^. Damage to cell membranes was assessed using the cell membrane thermostability test at temperatures of 40 °C and 45 °C, according with procedure described by Blum et al. (1981) [73]. Relative water content in leaf tissues was estimated through the methodology used by Sade et al. [74]. Results of relative water content were expressed in percentages.

### 4.5. Agronomic Variables

After occurrence of the vegetative period, number of days to flowering was measured by counting the number of days occurring from seeding until the day at which half of the panicle had their anthers exposed. At the end of flowering, measurements were collected from the base of the culm up to the tip of the panicle to measure plant height (in cm). Panicle length was measured from the base to the tip of the panicle (in cm). Harvest occurred on 1 of July (103 days after seeding) once the grain reached physiological maturity. Every panicle was shelled individually and grain weight per panicle was recorded (GWP, g). Additionally, the weight of 100 grains (W100G, g) and number of grains per panicle (NGP = 100 × (GWP/W100G)).

### 4.6. Statistical Analyses

Statistical analyses were performed using the statistical package InfoStat [75]. Analyses of variance (ANOVA) were performed for every response variable. When significant effects (*p* ≤ 0.05) were found, the mean comparisons were performed with Tukey tests (*p* ≤ 0.05). Physiological variables were composed of complete matrixes of data, hence total degrees of freedom were 95 ((two developmental stages x two soil humidity treatments x six grain sorghum genotypes × four reps) − 1). Agronomic variables, on the other hand, did not always were composed of complete matrixes [lost plots (plants), due to the lack of irrigation in the hydric stress treatments], thus degrees of freedom did not always account for 95 (Table 3). For the variables that were expressed in percentages, relative water content and damage to cell membranes (at 40 and 45 °C), an angular transformation was used (also known as arcsine square root or arcsine transformation) $\left[aresine{\left(Yi\right)}^{1/2}\right]$, and ANOVA and means comparisons were performed using the transformed values (*p* ≤ 0.05). The results of these variables were latter retransformed back to the original scale (percentages).

## 5. Conclusions

Hydric stress decreased the Fv/Fm and all physiological variables. Seed yield related variables were more affected by hydric stress pre-anthesis. Genotypes LER 1 and LER 2 were identified as tolerant to hydric stress, as they had lower damage to PSII; LER 1 and LEB 2 for their superior RWC; and LER 1 as a thermo tolerant genotype, due to its lower DCM at 45 °C. It is concluded that LER 1 could have the potential for hydric and heat stress tolerance in the arid and dry northeast Mexico.

## Figures and Tables

**Figure 1 plants-10-02265-f001:**
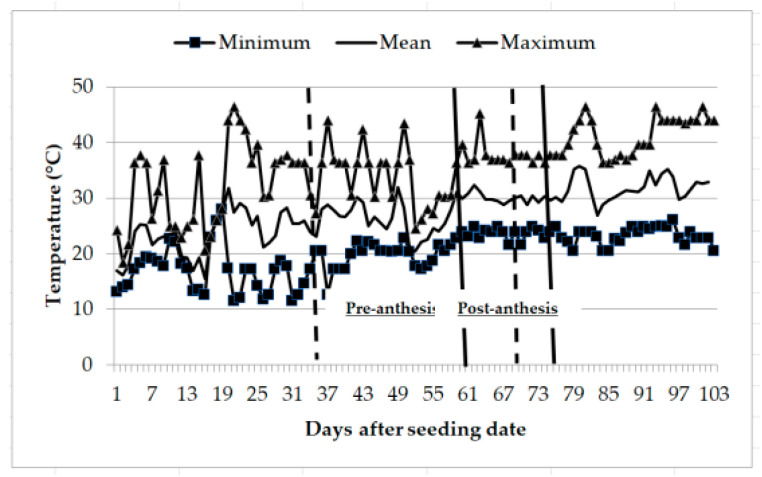
Minimum, mean, and maximum temperatures occurring through the duration of experiment conducted to identify tolerant genotypes to hydric and temperature stresses in northeast Mexico. Pre-anthesis occurred approximately between the two dotted lines while post-anthesis occurred approximately between the two solid lines.

**Figure 2 plants-10-02265-f002:**
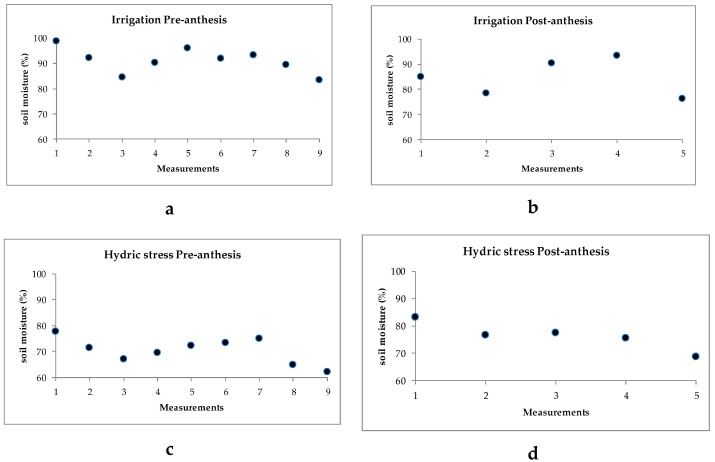
Soil humidity (%) under irrigation at pre-anthesis (**a**) and post-anthesis (**b**); and under hydric stress at pre-anthesis (**c**) and post-anthesis (**d**). Experiment conducted to identify tolerant genotypes to hydric and temperature stresses across samplings for irrigation and for hydric stress at both pre and post-anthesis in northeast Mexico.

**Figure 3 plants-10-02265-f003:**
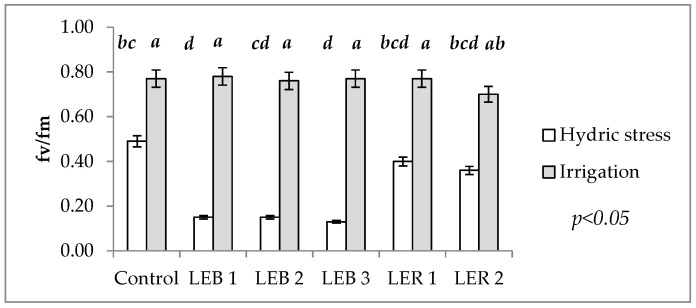
Effect of the soil humidity– genotype interaction on Fv/Fm in experiment conducted to identify tolerant genotypes to hydric and temperature stresses in northeast Mexico. Different letters above bars indicate significant (*p* < 0.05) differences among genotype and hydric condition treatment combinations.

**Figure 4 plants-10-02265-f004:**
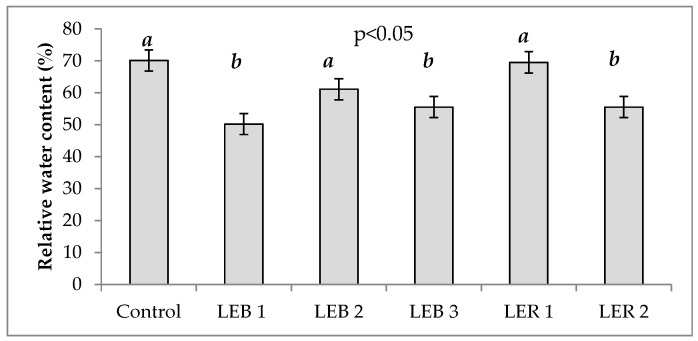
Relative water content of six genotypes averaged across two soil humidity treatments (hydric stress and irrigation) in experiment conducted to identify tolerant genotypes to hydric and temperature stresses in northeast Mexico. Different letters above bars indicate significant (*p* < 0.05) differences among genotypes.

**Figure 5 plants-10-02265-f005:**
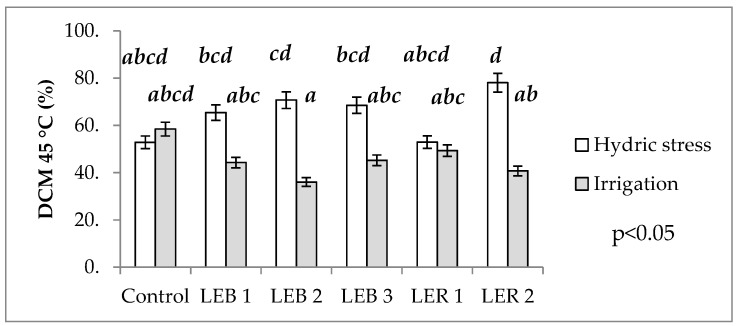
Effect of the soil humidity–genotype interaction on DCM at 45 °C in experiment conducted to identify tolerant genotypes to hydric and temperature stresses in northeast Mexico. Different letters above bars indicate significant (*p* < 0.05) differences among genotype and hydric condition treatment combinations.

**Figure 6 plants-10-02265-f006:**
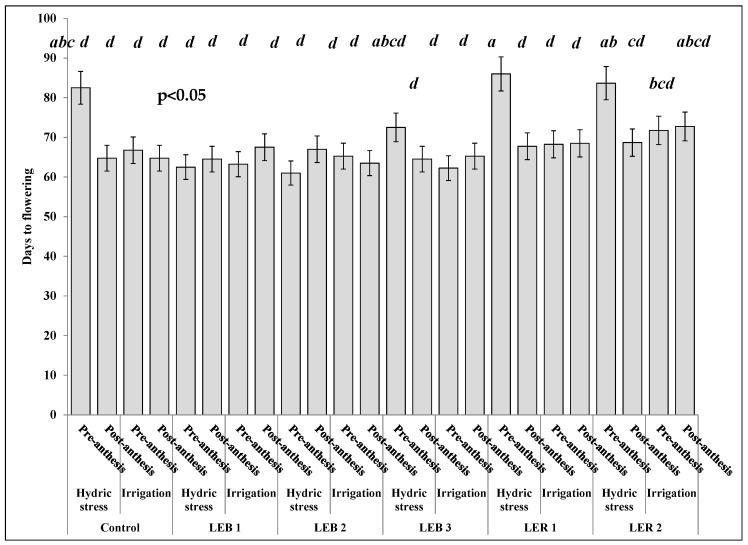
Effect of the soil humidity–development stage–genotype interaction on days to flowering in experiment conducted to identify tolerant genotypes to hydric and temperature stresses in northeast Mexico. Different letters above bars indicate significant (*p* < 0.05) differences among genotype, hydric condition, and sorghum development stage treatment combinations.

**Figure 7 plants-10-02265-f007:**
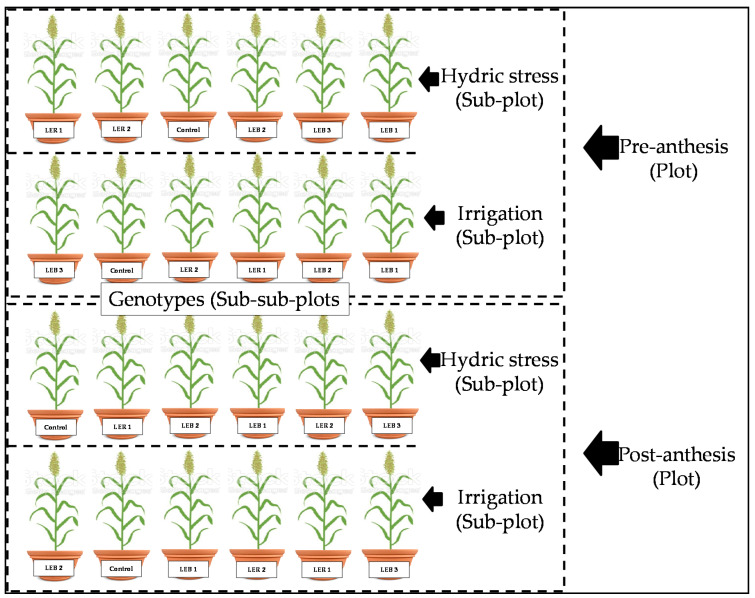
Representation of one (of four) of the replications in experiment conducted to identify tolerant genotypes to hydric and temperature stresses in Northeast, Mexico.

**Table 1 plants-10-02265-t001:** Mean squares, significance, and variation coefficient of physiological variables in experiment conducted to identify tolerant Genotypes to Hydric and Temperature Stresses in northeast Mexico.

S. V.	D.F.	Fv/Fm	P_N_ (μmol CO_2_ m^−2^ s^−1^)	g_s_ (mol H_2_O m^−2^ s^−1^)	E (mmol H_2_O m^−2^ s^−1^)	RWC (%)	DCM 40 (%)	DCM 45 (%)
Stage (S)	1	0.01 ns	18.2 ns	0.02 ns	24.7 *	45.7 ns	2737.1 ns	646.0 ns
Error (a)	6	0.02	18.2	0.02	2.7	153.3	708.6	203.3
Humidity (H)	1	5.47 **	12,248.2 **	1.83 **	1045.5 **	25,696.7 **	752.1 ns	7744.9 **
H x S	1	0.05 ns	69.5 ns	0.12 **	6.5 ns	19.3 ns	2377.5 ns	831.3 ns
Error (b)	6	0.02	15.9	0.01	1.4	115.9	513.4	182.4
Genotype (G)	5	0.09 *	14.1 ns	0.01 ns	3.4 ns	359.1 *	267.5 ns	121.8 ns
H x G	5	0.11 *	5.9 ns	0.01 ns	2.8 ns	300.24 ns	854.2 ns	1073.6 *
S x G	5	0.01 ns	20.6 ns	0.01 ns	2.4 ns	147.1 ns	261.0 ns	130.5 ns
H x S x G	5	0.01 ns	18.7 ns	0.01 ns	2.3 ns	158.3 ns	494.2 ns	279.9 ns
Error (c)	60	0.04	10.9	0.01	1.8	140.2	516.0	259.0
C.V.		38.8	35.5	60.6	27.8	23.2	71.3	29.3

Fv/Fm: Maximum quantum yield of PSII; P_N_: Photosynthetic rate; g_s_: H_2_O stomatal conductance; E: Transpiration rate; DCM 40: Damage to cellular membrane at 40 °C; DCM 60: Damage to cellular membrane at 45 °C; RWC: Relative water content; **: Significance at *p* ≤ 0.01; *: at *p* ≤ 0.05; ns: non-significant; and C.V.: Coefficient of variation.

**Table 2 plants-10-02265-t002:** Physiological parameter means with irrigation and under hydric stress in experiment conducted to identify tolerant genotypes to hydric stress and temperature stresses in northeast Mexico.

Variable	Hydric Stress	Irrigation
P_N_ (µmol CO_2_ m^−2^ s^−1^)	−1.9 b	20.6 a
g_s_ (mol H_2_O m^−2^ s^−1^)	0.06 b	0.34 a
E (mmol H_2_O m^−2^ s^−1^)	1.5 b	8.1 a
RWC (%)	34.8 b	67.5 a
DCM 40 (%)	32.6	24.9
DCM 45 (%)	81.2 a	51.4 b

P_N_: Photosynthetic rate; g_s_: Stomatal conductance; E: Transpiration rate; RWC: Relative water content; DCM 40: Damage to cellular membrane at 40 °C; and DCM 45: Damage to cellular membrane at 45 °C. Values with different letter within rows are significantly different (Tukey, *p* ≤ 0.05).

**Table 3 plants-10-02265-t003:** Mean squares, significance, and variation coefficient of agronomic variables in experiment conducted to identify tolerant Genotypes to Hydric and Temperature Stresses in northeast Mexico.

S. V.	d. f. †	Days to Flowering.	d. f.	Plant Ht. (cm)	d. f.	Panicle Lenght (cm)	d. f.	GWP (gr)	W100G (gr)	NGP
Stage (S)	1	130.8 ns	1	6750.3 **	1	968.7 **	1	2214.9 **	0.31 ns	1,684,749 ns
Error (a)	6	26.3	6	258.6	6	23.4	6	119.9	0.34	1,150,454.00
Humidity (H)	1	165.0 *	1	9861.8 **	1	709.9 **	1	20,009.5 **	6.09 **	40,694,690 **
H × S	1	383.3 **	1	7332.5 **	1	995.1 **	1	1846.9 **	0.64 *	1,296,639 ns
Error (b)	6	16.1	6	185.6	6	10.4	5	101.3	0.05	465,604
Genotype (G)	5	199.0 **	5	296.8 ns	5	39.9 *	5	1520.7 **	1.30 **	1,003,595 ns
H × G	5	36.3 ns	5	63.6 ns	5	8.2 ns	5	51.5 ns	0.49 **	288,298 ns
S × G	5	80.6 **	5	147.7 ns	5	19.9 ns	5	170.8 ns	0.22 *	350,376 ns
H × S × G	5	66.7 **	5	27.6 ns	5	16.4 ns	4	191.9 ns	0.16 ns	265,983 ns
Error (c)	50	19.6	60	192.5	55	15.5	26	295.5	0.09	872,887
C. V.		6.5		16.8		16.6		41.2	17.1	40.8

Plant Ht.: Plant height; GWP: Grain weight per panicle; W100G: Weight of 100 grains; and Number of grains per panicle. **: Significancia (*p* < 0.01); *: Significancia (*p* < 0.05); ns: no significativo; C. V.: Coefficient of variation. c † Degrees of freedom for the different agronomic variables are different from 95, since missing data in some of the variables. GWP, W100G, and NGP had the same dedrees of freedom.

**Table 4 plants-10-02265-t004:** Development stage–soil humidity interaction on agronomic variables in experiment conducted to identify tolerant Genotypes to Hydric and Temperature Stresses in northeast Mexico.

Stage	Soil Humidity	Days to Flowering		Plant ht. (cm)		Panicle Length (cm)		GWP (gr)		W100G (gr)		NGP	
Pre-anthesis	Hydric stress †	74	a	55.4	b	13.9	b	8.2	b	1.03	b	880	b
	Irrigation	66	b	93.1	a	26.2	a	61.6	a	2.04	a	3000	a
Post-anthesis	Hydric stress	66	b	89.6	a	27.4	a	16.2	b	1.32	b	1068	b
	Irrigation	67	b	92.4	a	26.5	a	46.5	a	1.81	a	2693	a

Plant Ht.: Plant height; GWP: Grain weight per panicle; W100G: Weight of 100 grains; and NGP: Number of grains per panicle. † Means with the same letters within rows are not signifficantly different (*p* = 0.05).

## Data Availability

The full database used for the present paper can be requested to Marisol Galicia-Juárez at her email address: marisol.galicia.juarez@uabc.edu.mx. Additionally, the authors are presenting the means of the main factors and all possible second order interactions for all the variables presented in the paper as an annex.

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
