# Peer review of "Identification of Sorghum (Sorghum bicolor (L.) Moench) Genotypes with Potential for Hydric and Heat Stress Tolerance in Northeastern Mexico"

_plants, 2021, doi:10.3390/plants10112265_

Round 1

Reviewer 1 Report

The manuscript reports the identitication of hydric and heat stress tolerance genotypes of sorghum through physiological and agronomical parameters, what has been achieved. The purpose of the work has been formulated correctly, but I could not find an explanation what is the originality and novelty of the research performed and the results obtained. Hypotheses should also be highlighted. My comments and suggestions are below.
Line 62 – It is the first explanation of the abbreviation (Fv/Fm).
Line 64 – PFI ?
Line 70-72 - what is the purpose of genotype selection?
Figure 2. - I suggest reducing the size of the graphs in Figure 2. I consider that the charts don't need to be that big and cover the whole page. Please consider also reducing the size of the other graphs.
Figure 3. - The differences in the RWC parameter were statistically significant for all 6 genotypes (under hydric stress compared to irrigation conditions) ? - not marked on the chart, the same for other charts.
Table 2. Please explain, why in the case of gas exchange parameters and DCM 40 (%) only mean values were shown, not values for individual genotypes? If these parameters were used to identify tolerant genotypes, it would be better to demonstrate their values for each genotype, the average seems to be insufficient.
Line 151 – ‘This difference represents a 40% difference.’ - I suggest rewriting.
Line 258-261 - Is the part of the text really refers to Figure 4?

In how many repetitions (biological, technical) each measurement was performed (Pn, gs, E, RWC, etc.)

Line 323 – ‘ELB’ – it should be LEB ?

Although I don't feel qualified to judge about the English language and style, however some linguistic and stylistic errors in the text are visible.

The reference list is up to date.

Author Response

Mexicali, Baja California, Mexico. 15 September, 2021

Dear Reviewers;

Attached to this, please review the new version of our paper. Additionally, we are submitting, as annex, all possible mean values for all three main factors as well as all second order interactions of all of our response variables.

Following we present two Tables. One for each of our two reviewers; were we punctually address each observation that was made to our paper.

We are convinced that we have made a very substantial improvement in the current version of our manuscript, as we earnestly worked with the best of our capabilities to correct every observation that was made.

We appreciate the kindness with which you addressed our faults/opportunities for improvement. The kindness and encouraging words made us taking the decision of resubmitting our manuscript.

Thank you very much.   

Dr. Jesús Santillano-Cázares

Profesor Universidad Autónoma de Baja California    

Mexicali, Baja California, México

jsantillano@uabc.edu.mx

Reviewer 1

Observation

Correction or clarification

I could not find an explanation what is the originality and novelty of the research performed and the results obtained

At the beginning of the writing process, the first authors of the paper had entitled the paper Maximum Quantum Yield of Photosystem II (PSII)…Believing that damage to PSII (Tv/Fm) was a variable that could impress the scientific community, particularly in Mexico, since the LI-COR 6400 equipment is scarce in Mexico.

When they asked my opinion, thinking about the Journal of Plants-MDPI, I knew that Fv/Fm would be anything but novel for readers of this journal. So I proposed to present the paper simply as an experimental lines testing study.

Nevertheless, the modest quota of novelty of our paper; what we are certain about is that drought and heat stress tolerant germplasm for all crops in the planet are going to have to be used, if we aspite to feed the growing population, and under current global warming conditions as well as water scarcity.   

Hypotheses should also be highlighted

The last paragraph of the introduction starts with our hypothesis: “The hypothesis of the present research was that there is at least one genotype that will shows tolerance to hydric and heat stress, from a group of promissory experimental lines that have been selected by two Mexican breeding programs”.

Line 62 – It is the first explanation of the abbreviation (Fv/Fm).

In the abstract, as well as in paragraph four in the introduction, the term maximum quantum yield of PSII (Fv/Fm) is written out in full and abbreviated.

After that, only the abbreviation (Fv/Fm) is used, for the most part.

Line 64 – PFI ?

Photosystem I has been written out in full.

Line 70-72 - what is the purpose of genotype selection?

The purpose of genotype selection was written out in the last paragraph of the introduction.

Figure 2. - I suggest reducing the size of the graphs in Figure 2. I consider that the charts don't need to be that big and cover the whole page. Please consider also reducing the size of the other graphs.

The size of all the Figures was reduced.

Figure 3. - The differences in the RWC parameter were statistically significant for all 6 genotypes (under hydric stress compared to irrigation conditions) ? - not marked on the chart, the same for other charts.

The graph about RWC in now figure 4 and literals have been added on top of each bar (genotype) for easy identification of significant differences among treatments. We have done the same for the rest of the graphs in the paper.

Table 2. Please explain, why in the case of gas exchange parameters and DCM 40 (%) only mean values were shown, not values for individual genotypes? If these parameters were used to identify tolerant genotypes, it would be better to demonstrate their values for each genotype; the average seems to be insufficient.

“Means in Table 2 are averaged over genotypes, since PN, gs or E weren´t affected by genotypes”. This is a sentence that we have added at the end of point 2.2.2. and again, one similar clarification) in the heading of Table 2.

Besides, this time we are including in the submission of our new version of our paper (as annex), a full list of means for every factor and 2nd order interactions, for all the response variables we are reporting; with the purpose of making completely transparent to readers all of our data from the present study.

Line 151 – ‘This difference represents a 40% difference.’ - I suggest rewriting.

At the end of paragraph about plant height (In point 2.3.2.) the observed sentence has been rewritten as follows:  “The difference between plant height under irrigation and at pre-anthesis was 40 % higher than plants under hydric stress at the same stage (pre-anthesis) (Table 4).”

Line 258-261 - Is the part of the text really refers to Figure 4?

No. We have fixed that. We wanted to refer to Figure 6.

In how many repetitions (biological, technical) each measurement was performed (Pn, gs, E, RWC, etc.)

Four repetitions for all physiological variables. In the case of some agronomical variables, we ended up with less than four reps, since we lost plots (single plants), particularly under the hydric stress treatment. This latter clarification was made in point 4.6, and again as a footnote in Table 3.

Line 323 – ‘ELB’ – it should be LEB ?

We have corrected that.

Although I don't feel qualified to judge about the English language and style, however some linguistic and stylistic errors in the text are visible.

In the current version we have made a careful review if our English writing and grammar. We hope the English part is acceptable in the current form.

Reviewer 2

Observation

Correction or clarification

On line 60 the word "sensitive" appears to have been swapped for "sensible".

The correction has been made.

Line 64: "PFI" is not previously defined.

Photosystem I has been written out in full.

Citation choices in the introduction tend to favor newer and more specialized application papers rather than foundational studies in the field, which does not reflect the explanation.  Expanding the introduction to address these narrow topics or using landmark papers which serve as useful background for the broad themes of the field as references would be more valuable to readers.

In the current version of our paper we have included citations using the criteria of picking only the papers with the most citations; although these papers are now old, for the standards of many journals (questionable requirement of using only very new literature). Thus, we are now citing highly referenced papers (classic), along with new ones.

There are linear trendlines in Figure 2 for what are obviously sigmoid trends.  The value of these trendlines should be explained or they should be removed.

We have deleted the trend lines. There was no point for the paper to use them.

Table 1 is generally unclear due to having multiple overlapping parameters and units.  Some column abbreviations are not defined, additionally.

Table 1 has been fixed. The respective units have been written out for each variable in Tables 1 (agronomic variables) and Table 3 (agronomic variables).

This manuscript appears to have been written with the Materials and Methods section before the results and then reorganized.  The Results section repeatedly references abbreviations that are introduced in the Materials and Methods section without prior explanation otherwise. 

Your perception is correct. In the first drafts of our paper, we had Materials and Methods, Results, and Discussion. In the current version we are following the Journal template. We have been careful to write out in full, followed by the abbreviation the first time each of the abbreviations are used for the first time in the paper.

This manuscript essentially cannot be read straight through because it is not clear what the genotypes being discussed in Results are without previously reading Materials and Methods.  It is necessary to either reorder the sections or explain what the organisms of study are earlier in the text.

We believe we have been taken care of providing the most information in the Introduction and Results without invading the Materials and Methods section.

We have modified the last paragraph in the introduction to describe the genotypes we used in the study, so reader can read the paper without having to jump across sections.

"a" and "b" in Table 2 are not clear- are these groupings?

Values with different letter within rows in Table 2 are significantly different. The two columns are being compared (Hydric stress versus Irrigation).

Line 302 - "masl" should be written out.

“meters above the sea level” has been written out in full in point 4.1.

Were plants supplied with adequate nutrients (N, P, S, etc.) for the duration of experimental growth?  This is not particularly clear from the methods section wording.

Besides the nutrient content of the soil used “[3.9% organic matter and 0.35 % nitrogen (N), pH of 7.9]. No other nutrient application was made throughout de duration of the experiment”. This sentence was included in point 4.2.

There are random breaks between sections- this may be a formatting issue at various stages in submission but it's worth checking.

We have left no breaks in the current version of our manuscript this time.

Author Contributions still contains default text.

Author Contributions have been properly described now.

Reviewer 2 Report

This is a conceptually solid manuscript, and the authors' work is commendable.  However, the presentation is significantly unclear, as detailed below.

English review is necessary for spelling, grammar, and word choice; as an example, on line 60 the word "sensitive" appears to have been swapped for "sensible".

Line 64: "PFI" is not previously defined.

Citation choices in the introduction tend to favor newer and more specialized application papers rather than foundational studies in the field, which does not reflect the explanation.  Expanding the introduction to address these narrow topics or using landmark papers which serve as useful background for the broad themes of the field as references would be more valuable to readers.

There are linear trendlines in Figure 2 for what are obviously sigmoid trends.  The value of these trendlines should be explained or they should be removed.

Table 1 is generally unclear due to having multiple overlapping parameters and units.  Some column abbreviations are not defined, additionally.

This manuscript appears to have been written with the Materials and Methods section before the results and then reorganized.  The Results section repeatedly references abbreviations that are introduced in the Materials and Methods section without prior explanation otherwise.  This manuscript essentially cannot be read straight through because it is not clear what the genotypes being discussed in Results are without previously reading Materials and Methods.  It is necessary to either reorder the sections or explain what the organisms of study are earlier in the text.

"a" and "b" in Table 2 are not clear- are these groupings?

Line 302 - "masl" should be written out.

Were plants supplied with adequate nutrients (N, P, S, etc.) for the duration of experimental growth?  This is not particularly clear from the methods section wording.

There are random breaks between sections- this may be a formatting issue at various stages in submission but it's worth checking.

Author Contributions still contains default text.

Author Response

(The authors gave the same response as above.)

Round 2

Reviewer 1 Report

The manuscript has been improved.

Author Response

Dear Reviewer:

Thank you.

This manuscript is a resubmission of an earlier submission. The following is a list of the peer review reports and author responses from that submission.

Round 1

Reviewer 2 Report

The experiment conducted by the authors is interesting and provides quite valuable material. However, the data collected have not yet been fully analyzed. For example, there is no comparison of physiological parameters at different stages of development, and the information in Table 2 raises the question of which stage of development measurements are presented here? There are more similar uncertainties. In addition, it would certainly be more informative for the reader to read the results first and then read the discussion in a separate section. Moreover, the journal's instructions to the authors state that the sections of the results and discussion must be separate.

The current section of the discussion is weak, lacking a deeper analysis of the causality of the changes obtained, such as the link between physiological and yield indicators, etc.